# Distinct Alterations in Dendritic Spine Morphology in the Absence of β-Neurexins

**DOI:** 10.3390/ijms25021285

**Published:** 2024-01-20

**Authors:** Leonie Mohrmann, Jochen Seebach, Markus Missler, Astrid Rohlmann

**Affiliations:** Institute of Anatomy and Molecular Neurobiology, University Münster, 48149 Münster, Germany; lmohrman@uni-muenster.de (L.M.); seebach@uni-muenster.de (J.S.)

**Keywords:** cell adhesion molecules, synaptic plasticity, mushroom spine, 3D reconstruction, perforated PSD, transmission electron microscopy, dense-core vesicles, mouse, hippocampus

## Abstract

Dendritic spines are essential for synaptic function because they constitute the postsynaptic compartment of the neurons that receives the most excitatory input. The extracellularly shorter variant of the presynaptic cell adhesion molecules neurexins, β-neurexin, has been implicated in various aspects of synaptic function, including neurotransmitter release. However, its role in developing or stabilizing dendritic spines as fundamental computational units of excitatory synapses has remained unclear. Here, we show through morphological analysis that the deletion of β-neurexins in hippocampal neurons in vitro and in hippocampal tissue in vivo affects presynaptic dense-core vesicles, as hypothesized earlier, and, unexpectedly, alters the postsynaptic spine structure. Specifically, we observed that the absence of β-neurexins led to an increase in filopodial-like protrusions in vitro and more mature mushroom-type spines in the CA1 region of adult knockout mice. In addition, the deletion of β-neurexins caused alterations in the spine head dimension and an increase in spines with perforations of their postsynaptic density but no changes in the overall number of spines or synapses. Our results indicate that presynaptic β-neurexins play a role across the synaptic cleft, possibly by aligning with postsynaptic binding partners and glutamate receptors via transsynaptic columns.

## 1. Introduction

Dendritic spines (DSs) are minute protrusions that extend laterally from neuronal dendrites and receive most of the excitatory input in brain circuitries [1,2,3]. DSs play a pivotal role in neurotransmission because they constitute a functionally distinct compartment of the postsynaptic neuron that integrates presynaptic inputs and mediates learning algorithms [2,4,5]. Mature, stable DSs typically display a mushroom-shaped morphology and increase in abundance with age [6,7,8]. The relatively large, bulbous head of a mushroom spine is linked to its dendrite by a thinner neck and contains a postsynaptic density, signaling molecules, actin-based scaffolds, and organelles such as endosomes or the spine apparatus [3,9,10,11]. Since spines undergo adaptive changes in response to stimuli, it was concluded that spines’ morphology and function are mutually dependent [12,13,14]. For example, the normally round spine head has a flat surface that faces the presynaptic terminal and can assume a concave morphology following plasticity-inducing signals [15]. Conversely, dysfunction can also manifest in structural changes because numerous neuropsychiatric disorders and behavioral deficits have been linked to the impairment of different types of spines and their morphologies [10,16,17,18].

Neurexins (Nrxn) are a large family of essential presynaptic cell adhesion molecules that regulate various aspects of synaptic function [19,20,21]. All three *Nrxn* genes in vertebrates contain independent promoters that drive the transcription of structurally larger α-neurexins (*αNrxn*) and smaller β-neurexins (*βNrxn*), the latter differing due to the expression of a β-specific 37-residue-long domain before splicing into the last (sixth) laminin–neurexin–sex-hormone-binding globulin (LNS) domain of the respective gene [20,21], with more variants arising from up to six conserved alternative splice sites [22,23]. Since Nrxn engages in binding activities with several postsynaptic partners and is part of transsynaptic nanocolumns [24,25], we expected that it may affect the stabilization of the DS structure [26]. However, previous work in constitutive or conditional knockout mice of multiple Nrxn variants reported normal overall spine density [27,28,29,30,31].

It has remained a conundrum if and how the major α- and βNrxn variants differ in their function at synapses. Although they share numerous binding partners, such as neuroligins (Nlgns) [32,33,34], leucine-rich repeat transmembrane neuronal (LRRTM) proteins [35,36,37], α-dystroglycan [38,39], latrophilins [40], and cerebellins (Cblns) [41,42,43], α- and βNrxn differ in the degree of phenotypic alterations when their functions are probed in deletion mouse models: The combined knockout of all three *αNrxn* genes is perinatally lethal and was mechanistically traced back to a strong impairment of Ca^2+^-dependent release of synaptic vesicles from excitatory and inhibitory synapses [29,44,45,46]. The selective deletion of all *βNrxn*, in turn, does not significantly impair survival and displays only moderately reduced excitatory release, as well as diminished presynaptic Ca^2+^ transients and dense-core vesicles (DCVs) [27,31,47]. We did, however, notice putative changes in dendritic protrusions in the absence of βNrxn protein variants during our recent investigation of neuromodulator-containing DCVs [47], prompting this current study to re-analyze two βNrxn deletion models for an altered spine morphology.

Here, we combined neuronal cell culture analysis and transmission electron microscopy of the brain tissue of conditional and constitutive knockout mice of βNrxn to explore the structure of dendritic spines in detail. Our findings indicate that βNrxn not only regulates Ca^2+^-dependent synaptic transmission but may also serve a role in the differentiation or stabilization of the normal dendritic spine structure.

## 2. Results

### 2.1. DCV Distribution Is Altered in Cultured Neurons of Two βNrxn Deletion Mouse Models

Endogenous βNrxns are highly mobile molecules that are mostly on axons and within synaptic terminals [31], where they affect not only the release of classical synaptic vesicles [27,31] but also the number of DCVs [47]. Strikingly, neuromodulators such as brain-derived neurotrophic factor (BDNF) are secreted from presynaptic DCVs [48,49,50] and play a major role in spine development and plasticity [51,52]. Since replicability concerning animal models and experimental conditions is a topic of great concern in general and in this journal [47], we decided to study DCV numbers in two different β-Nrxn deletion models. To ensure that the results from conditional and constitutive knockout of βNrxn variants were comparable, we first tested our previous conclusion of reduced numbers of DCVs [47] in both βNrxn knockouts.

We conditionally deleted floxed βNrxn-specific exons by using Cre recombinase-expressing lentivirus to remove all βNrxn variants in primary hippocampal neurons from KI mice, as described before [31,47]. Inactive Cre recombinase (Cre^mut^)-expressing KI neurons served as controls, and efficient reduction of βNrxn was validated with immunoblots, as shown in [31]. Similarly, we cultured hippocampal neurons from a constitutive βNrxn TKO mouse line generated through germline deletion and controls [31,47]. To probe the distribution of DCVs in both types of deletion models, we immunolabeled the neuronal cultures at DIV19 with antibodies against chromogranin A (ChrgA), a common matrix protein of DCVs present in a subset of axons [53,54,55]. We observed ChrgA-positive clusters in axons of control neurons, for example, traveling along a neighbored neuronal soma (Figure 1A) and less abundantly on axons of βNrxn TKO neurons (Figure 1B), thereby closely resembling the images of Cre^mut^ and Cre-expressing βNrxn KI neurons from our earlier study of DCVs [47]. Measurements of the intensity of ChrgA fluorescence within a defined axonal window revealed a 35% reduction in ChrgA fluorescence intensity in axons of βNrxn cKO neurons compared to control neurons in our current set of experiments (Figure 1C; Cre^mut^ control: 7846 ± 442 arbitrary units [A.U.], Cre cKO: 5133 ± 531, *p* = 0.0003), an even stronger effect than that reported before [47], which was presumably due to the greater number of experiments in this study. In cultures of the constitutive βNrxn KO model, the ChrgA fluorescence intensity was less strongly but also significantly diminished by 17% (Figure 1D; WT control: 7132 ± 340 arbitrary units [A.U.], TKO: 5949 ± 320, *p* = 0.023). Together, these results confirm our previous conclusion that βNrxn is required for normal levels of ChrgA-containing DCVs in hippocampal neurons [47]. Importantly, they also indicate that phenotypic observations made upon conditional deletion can be reproduced in constitutive βNrxn TKO mutants, despite the caveats associated with potential compensatory effects in germline deletions [21].

### 2.2. Dendritic Spine Alterations in Cultured βNrxn-Deficient Neurons

During our analysis of neuromodulator-containing DCVs, we noticed a large number of longer dendritic protrusions in the absence of βNrxn in hippocampal neurons, which presumably corresponded to filopodia [1,2,3]. A previous study using cultured neurons derived from the cerebral cortex did not report such defects [27]. We first tested if the overall density of dendritic protrusions was changed in our primary neurons cultured from the hippocampi of βNrxn mutants. To visualize dendrites and their protrusions (classical spines and filopodia), we expressed cytosolic t-dimer-RFP in control (Figure 2A,B) and knockout (Figure 2C) neurons. We then determined their density by normalizing absolute numbers of protrusions along randomly chosen dendritic segments to the length of these segments. In cultures of the constitutive βNrxn TKO model, the overall density was undistinguishable from that in the WT control (Figure 3A; WT control: 6.49 ± 0.19 protrusions/10 μm dendrite length, TKO: 6.96 ± 0.16, *p* = 0.053), which was consistent with the earlier results in cortical neurons [27]. Attesting to the reliability of our approach, we observed very similar values in measurements from cultures of βNrxn KI neurons expressing Cre recombinase (cKO) or inactive Cre^mut^ for the control (Cre^mut^ control: 6.62 ± 0.20 protrusions/10 μm, Cre cKO: 6.82 ± 0.21, *p* = 0.48). Thus, the deletion of βNrxn in cultures from the hippocampus had no impact on the overall density of dendritic protrusions.

In our analysis of RFP-transfected neurons from hippocampal cultures, however, we realized that many protrusions from βNrxn-deficient cells extended farther from their dendrites than in comparable controls (Figure 2B,C). To quantify this effect, we measured the length and classified all protrusions as either longer or shorter than 2 μm (Figure 2B,C). Confirming the initial observation, we found that the number of filopodia (protrusions > 2 μm) was increased by almost 40% in dendrites of constitutive βNrxn TKO neurons compared to the controls (Figure 3B; WT control: 2.99 ± 0.13 protrusions/10 μm, TKO: 4.13 ± 0.13, *p* < 0.0001). Conversely, protrusions below 2 μm in length were reduced by almost 20% (Figure 3C; WT control: 3.50 ± 0.13 protrusions/10 μm, TKO: 2.83 ± 0.11, *p* = 0.0002). We did not try to further distinguish between dendritic spine types (stubby, thin/long, mushroom). In addition, the βNrxn-deficient neurons not only had a higher number of longer protrusions but also revealed a higher number of branched protrusions, which was increased by 45% in comparison to the controls (Figure 3D; WT control: 0.82 ± 0.06 protrusions/10 μm, TKO: 1.19 ± 0.07, *p* < 0.0001). While this difference appears to be considerable, the absolute number of branched spines amounted to only 13% in the controls and 17% in the TKO neurons. Together, these data show that dendritic protrusions in culture extended farther and branched more frequently in the absence of all βNrxn isoforms, while their overall number remained normal. This increase in filopodia could reflect the homeostatic spinogenesis that occurs in mature neurons when synaptic transmission is reduced [56], as is the case in βNrxn mutants [27,47].

### 2.3. Dendritic Spine Alterations in the Hippocampal Tissue of βNrxn-Deficient Mice

Our observation of shifts between subpopulations of dendritic protrusions, i.e., shorter vs. longer and branched vs. unbranched, indicated that ßNrxn may exert a subtle but significant influence on spine morphology. It is important to note that even small changes—for example, in spine neck length or head width—may represent major functional differences [57,58]. To investigate such morphological alterations in more detail, we decided to focus our further analysis on hippocampal brain tissue from constitutive ßNrxn TKO mice. The rationale for this restriction was fourfold: First, analyses in culture through conventional fluorescence microscopy tend to underestimate distinct subtypes of spines [57]; second, the geometry of the extracellular space in intact tissue affects spinous synapses [59]; third, the physical contact with perisynaptic astrocytic processes of glia cells, which were absent in our sandwich cell culture [60], is known to affect the dendritic spine structure [61,62,63]; fourth, transmission electron microscopy of fixed tissue samples still represents the benchmark for exquisitely detailed and quantitative analyses of spine morphology [11].

To compare putative changes in spine morphology in intact brain tissue with our results from cultured primary hippocampal neurons (Figure 2 and Figure 3), we prepared samples of the hippocampal CA1 region for electron microscopy and analyzed images from the stratum radiatum (Figure 4A,B) because most excitatory inputs terminate on spines in this layer [11].

We first quantified the overall number of presumptive excitatory synapses by determining the area density of asymmetric terminals based on the classical criteria for ultrastructurally defined type 1 synapses [64], as also described in Section 4.3.3. Consistently with the unchanged glutamatergic synapse density in cortical cultures [27], we found that the area density of type 1 terminals was very similar in WT control and ßNrxn TKO mice (Figure 4C; WT control: 25.3 ± 0.79 type 1 contacts/45 μm^2^, TKO: 25.7 ± 0.93, *p* = 0.7741). Our cell culture analysis of dendritic protrusions revealed an increase in filopodia (Figure 2 and Figure 3) but did not address the most mature and stable subtype of dendritic spines in the stratum radiatum of the adult CA1 hippocampus [6,7,8]. Therefore, we outlined mushroom spines on the electron microscopic images (Figure 4A,B; mushroom spines colored in red), as defined in Section 4.3.3. Visual appearance, as well as quantification of the area density of mushroom spines, confirmed an increase of almost 50% in the absence of ßNrxn (Figure 4D; WT control: 12.5 ± 0.92 spines/345 μm^2^, TKO: 18.5 ± 0.99, *p* < 0.0001), indicating that deletion of the synaptic cell adhesion molecule ßNrxn affected the number of the most mature subpopulation of spines.

To investigate whether ßNrxn also altered the ultrastructural morphology of mushroom spines, we next reconstructed about 60 samples from serial sections of control and TKO hippocampi in 3D. Suitable viewing angles were selected, and 10 representative images of 2D renderings from each genotype are shown in Figure 5. To focus on the heterogeneity of their ultrastructure, we adjusted the scales to display the samples at comparable sizes. Although the spines displayed were from the same brain region and belonged to the same subpopulation, they were all variable in shape and revealed a unique morphology (Figure 5A,B), as noted before in comparable studies using 3D reconstructions, particularly those of hippocampal neurons [11,65,66].

While most morphological features, such as curved and straight necks or round and oval heads, were detectable in both the control and ßNrxn TKO samples, we noticed a possibly higher number of spines with a prominent perforated PSD in ßNrxn-deficient reconstructions (arrows in Figure 5A,B). To test this hypothesis, we identified dendritic spine profiles with a perforated PSD in higher-resolution electron microscopic images (Figure 6A,B) and quantified their area density in ßNrxn TKO samples; this was slightly but significantly elevated by about 28% compared to the controls (Figure 6C; WT control: 4.6 ± 0.4 perforated PSDs/345 μm^2^, TKO: 5.85 ± 0.44, *p* = 0.037). This increase in dendritic spines containing a perforated PSD (Figure 6B) was consistent with the elevated number of mushroom-type protrusions in the ßNrxn-deficient samples (Figure 4D) because mushroom spines are generally more likely to contain perforated PSDs [67]. In addition, we noticed dendritic spines with other characteristic ultrastructural features, such as a spine apparatus (Figure 6D) or spinules (Figure 5), but preliminary estimations did not suggest quantitative differences between the genotypes analyzed here.

Alterations in the functional strength or responsiveness of spinous synapses are often reflected by changes in the ultrastructure of PSDs [68], such as the increase of perforated PSDs shown here (Figure 6). Additional ultrastructural modifications are known to reflect dendritic spine plasticity—for example, spine head size [65,69] or neck dimensions [57,70]. To finally investigate if these structures were affected by the deletion of the ßNrxn variants, we outlined the head and neck of mushroom spines in our electron microscopic images (Figure 7A). Quantitative analysis of the head area of mushroom spines revealed a small reduction of about 12% in ßNrxn-deficient mice compared to the controls (WT control: 0.17 ± 0.008 μm^2^, TKO: 0.15 ± 0.008 μm^2^, *p* = 0.0254), whereas their neck length remained unchanged (WT control: 0.32 ± 0.012 μm, TKO: 0.33 ± 0.015 μm^2^, *p* = 0.5926). Together, our morphological results demonstrated that the deletion of ßNrxn did not change the overall number of excitatory (asymmetric) synaptic contacts but caused a shift towards more mushroom-type spinous synapses with a slightly smaller head size and a greater proportion of perforated PSDs.

## 3. Discussion

Studies using so-called chimeric synapse formation assays in vitro demonstrated a strong synaptogenic activity for recombinantly expressed ßNrxn [32,71,72,73,74], which fostered the expectation that the deletion of these molecules would reduce the number of all or at least of a subpopulation of synaptic contacts. Contrary to such expectations, we showed here in ßNrxn knockout mice that the overall number of asymmetric synapses and dendritic protrusions remained unchanged. Strikingly, the absence of ßNrxn even led to an increase in longer filopodial-like protrusions and more mature mushroom-type spines with a slightly reduced head size, as well as more spines with perforations of their PSDs. Thus, we suggest that presynaptic ßNrxn is necessary to maintain normal proportions of spine subpopulations and a normal dendritic spine structure.

### 3.1. Our Strategy for Detecting the ßNrxn Spine Phenotype Is Sensitive and Reliable

We hypothesized based on previous analyses of ßNrxn knockouts [27,31,47] that any putative morphological phenotype at synapses, if present at all, was likely moderate rather than bold. Therefore, we strictly observed the following precautions to obtain reliable results: (i) The investigators were blinded to the identity of genotypes of the samples; (ii) all experiments were carried out in at least three independent biological replicates, i.e., three different brains/three mice and three culture preparations at different timepoints from offspring of three timed pregnancies; (iii) the major conclusions were based on at least two different experimental strategies or distinct methods, i.e., cultures/in vitro versus tissue/in vivo and conditional versus constitutive knockout neurons; (iv) we made sure to analyze a large enough number of spines (>3000), dendritic length (>5000 μm), or tissue area (>13,000 μm^2^) per genotype, with all exceeding the recommendations for samples sizes recently outlined in this journal [10].

We first investigated dendritic filopodia and spines in primary neurons in vitro due to the superior visibility of fluorescently labeled structures, such as filopodia, spines, or DCVs, in cultures when using light microscopy. An advantage of primary neurons grown at low density on glass coverslips is that fewer protrusions are hidden from view due to their position on the dendrite. Attesting to the robustness of this approach, we determined a density of 6.6 protrusions/10 μm dendritic length in lentivirus-treated control neurons (Cre^mut^) from the conditional ßNrxn mouse line, which was almost identical to the 6.5 dendritic protrusions/10 μm measured in control neurons of the constitutive knockout line (Figure 3). These values are also similar across studies, as Anderson and colleagues found 7.5 spines/10 μm in cultured cortical control neurons from the same cKO line as that used here [27]. Furthermore, our values fit into the general range of 2–10 spines/10 μm dendrite depending on the area of origin and age of neurons, as well as on the use of dissociated culture or intact tissue experiments [75,76]. On the downside, neuronal cultures tend to show an inevitable degree of variability depending on the conditions, such as plus/minus astrocytes, plus/minus serum, or the cell density chosen; these are all factors that might affect the time course of spinogenesis and the density of spines. Moreover, due to the inherent resolution limits of conventional epifluorescent microscopy, any classification of spines that relies on precise estimations of head and neck diameters is likely inaccurate [77]. In particular, recent studies using super-resolution microscopy have cautioned that short spines are not sufficiently resolved, leading to an overrepresentation of so-called stubby spines with a limited resolution [57]. 

Therefore, in the culture model, we measured the labeled DCVs and density of RFP-filled filopodial-like or spinous protrusions by using fluorescent microscopy (Figure 1, Figure 2 and Figure 3), but we refrained from a more detailed quantitative investigation of the mushroom-type subpopulation of spines, the prototypical mature and stable subtype in the stratum radiatum of adult CA1 hippocampus [6,7,8]. For this, we relied on fixed tissue from the corresponding hippocampal area and transmission electron microscopy (Figure 4, Figure 6, and Figure 7) because EM remains the gold standard for the unbiased investigation of the precise dimensions of synapse morphology [78]. Additionally, EM enables three-dimensional reconstructions from serial sections to explore individual protrusions at the highest possible resolution (Figure 5). Thus, we conclude that our approach in this study produced reliable results and was sensitive enough to detect small to moderate differences in spine morphology.

### 3.2. Putative Mechanisms for Explaining the ßNrxn Spine Phenotype

ßNrxn variants are less abundant than αNrxn variants [79,80] and show distinct dynamic behavior at the cell surface of axons and synapses, where ßNrxn molecules appear less mobile [31,81,82]. Based on their total protein amount in the lowest femtomolar range per µg of brain tissue, the copy number of ßNrxn proteins was estimated at 7–16 molecules per synapse [79]. These values might yet be an overestimation, since we recently found by using uPAINT, dSTORM, and immunoelectron microscopy that about 50% of endogenous ßNrxn molecules are present on the axon outside synapses and that less than 40% of glutamatergic terminals in hippocampal neurons contain ßNrxn [31]. These data suggest that very few copies of ßNrxn might be responsible for the remarkable regulation of both presynaptic DCVs and postsynaptic structures reported here. The phenotype of an elevated density of mature mushroom-type spines with a slightly reduced head size and more synapses with perforations of their PSDs may, in fact, point to a rather specific role of ßNrxn in the regulation of spinous synapses. Such a specific role underscores the earlier idea that βNrxn variants also have non-redundant functions with respect to those of αNrxn [46], and, consistently, no alterations in spine structure were found in KO mice lacking multiple αNrxns [30]. How, then, can the spine phenotype in KO mice of these highly elusive presynaptic molecules be explained? Possible explanations are that (i) alterations could be a consequence of reduced presynaptic spontaneous and evoked release of synaptic vesicles, (ii) alterations could be a consequence of reduced dense-core vesicle release, (iii) alterations could originate from a missing link in a transsynaptic signaling chain that originates from presynaptic β-Nrxn and targets postsynaptic receptors/scaffolding molecules, or (iv) they could be due to any combination of these effects.

A straightforward possibility would be to point out that the spine formations or their subtype ratios depend on intact excitatory synaptic release [83], and ßNrxn KO neurons suffer from decreased miniature EPSC frequency, lower presynaptic Ca^2+^ influx, and reduced AMPAR- and NMDAR-mediated evoked EPSCs, as reported previously [27,31,47]. However, we did not observe reduced spine numbers but, rather, a specific shift towards more mature mushroom-type spines in adult mouse brains, arguing against this possibility. In addition, our result is consistent with those of other mouse models that completely lacked spontaneous and action-potential-evoked presynaptic transmitter release from excitatory glutamatergic synapses but developed an almost normal spine architecture [84,85,86]. In a similar argument, one could suspect that the lower number of neuromodulator-containing DCVs in ßNrxn-deficient neurons (Figure 1 and [47]) is responsible for the spine phenotype. Neuromodulators such as brain-derived neurotrophic factor (BDNF) are stored in presynaptic DCVs of hippocampal neurons [48] and stimulate dendritic spine growth and maturation independent of synaptic activity [87,88]. It was a remarkable finding in the earlier analysis that the deletion of the very few copies of ßNrxn was sufficient to shift a considerable proportion of presynaptic terminals containing 1–2 DCVs to terminals with zero DCVs [47]. In line with these data, reduced DCV/BDNF levels would rather explain a reduced density of mature mushroom spines, which was not found in this study (Figure 4). Thus, explanations based on the presynaptic functions of ßNrxn in neurotransmitter or neuromodulator release appear unlikely to account for their role in regulating spine structure.

An alternative possibility to explain how presynaptic ßNrxn might regulate spine plasticity could emphasize that Nrxn molecules constitute building blocks of transsynaptic nanocolumns [25,89]. According to this concept, the different postsynaptic binding partners of Nrxn at excitatory synapses—for example, Nlgn, LRRTM, or Cbln—might preferentially connect them to particular glutamatergic postsynaptic receptors [25,89] that are arranged in distinct clusters within the postsynaptic density [90,91]. For example, splice variants of Nrxn1 have been shown to control NMDA receptor responses, and Nrxn3 isoforms control AMPA receptor strength [26,92,93,94]. Since AMPAR-to-NMDAR ratios are linked to spine structure [95,96,97], alterations in spine morphology should be no surprise in the absence of distinct Nrxn variants, as shown in this study for ßNrxn. Although this concept appears stringent, there are also confounding observations from knockouts of the postsynaptic binding partners of Nrxn. For example, deletion models of multiple Nlgn variants revealed no major effects on spine numbers [98,99,100], whereas the ablation of another prominent partner, LRRTM1, alone and in combination with SynCAM1, produced a clear reduction in spine numbers [101]. Moreover, the deletion of LRRTM1 led to an altered spine morphology with longer spines [102], resembling our finding in neuronal cultures (Figure 2 and Figure 3), and a shift between different subtypes toward fewer mushroom spines [101], the opposite effect of what we observed in βNrxn KOs (Figure 4).

The concept that Nrxn at excitatory synapses may impact AMPAR and NMDAR signaling through alignment in transsynaptic nanocolumns could reconcile the seemingly disparate presynaptic and postsynaptic effects seen in βNrxn mutants here and elsewhere [27,31,47]. Postsynaptic spine plasticity accompanying learning and memory processes can be triggered by sensory stimulation or local glutamate signaling, which requires the opening of NMDAR [103,104]. Moreover, transient changes in synaptic strength produce opposing effects, as spines expand during long-term potentiation (LTP) and shrink with long-term depression (LTD) [8,76]. Accordingly, the reduced spine head dimension reported here for ßNrxn KO neurons (Figure 7) could reflect such an adaptive change because the deletion of ßNrxn blocked the induction of LTP in neuronal subpopulations [27]. In support, these adaptive changes often involve actin remodeling in spines [1], and actin-dependent adaptations of spine head curvature could be blocked by the exogenous addition of soluble recombinant Nrxn1ß, possibly by disrupting the formation of transsynaptic complexes with Nlgn [15]. Together, the available data suggest that βNrxn, along with its binding partners and postsynaptic receptors, is part of transsynaptic nanocolumns, which constitute signaling pathways that orchestrate pre- and postsynaptic function and structure.

### 3.3. Disease Associations of the ßNrxn Phenotype

Human genetic studies have linked disruptions in all three human Nrxn genes (NRXN1-3) to neurodevelopmental disorders such as autism spectrum disorders (ASDs) and schizophrenia (SCZ) [105,106,107,108]. Although these disorders present complex genetic pictures consistent with a scenario with multiple rare variants and a large number of candidate genes [109,110], defects in Nrxn genes are among the most frequently found variants in ASD cohort studies [111,112]. According to ClinVar, there are currently close to 2000 mutations in the Nrxn genes identified. Most cases represent copy-number variations in parts of or the entire gene [113,114,115,116,117]. These clinical studies have also provided valuable insights into a remarkable functional pleiotropy of Nrxn molecules because individuals with ASDs, SCZ, intellectual disabilities, or epilepsy can all harbor Nrxn mutations [106,108]. This pleiotropy, along with a frequently observed incomplete penetrance, may point to complex genetic compensatory mechanisms at the transcriptomic level [118,119]. Importantly, the pathogenic mutations overlapping with Nrxn are exonic, rare, and non-recurrent, and the majority of patients harbor deletions [117], providing a rationale for the investigation of deletion mouse models.

Even though ßNrxn-specific genomic sequences cover only a small fraction of the chromosomal areas of their respective genes [23,108], several ASD-associated cases have been linked to these ßNrxn sequences. Two rare missense mutations have been identified in exon 18, the first coding exon of Nrxn1ß (amino acid changes: S14L and T40S), causing abnormalities in its unusually long signal peptide [120]. Another study in a different cohort of cases identified two ßNrxn-specific mutations in the Kozak sequence and the initiator methionine of Nrxn1ß in patients suffering from ASD and mental retardation, leading to reduced levels of Nrxn1ß at synapses [121]. Most recently, a case study of a young girl with ASD and global developmental delay reported a ß-specific frameshift mutation 15 amino acids downstream of the signal peptide in exon 18 of Nrxn3 [122]. In addition, some patients with deletions covering sequences shared by αNrxn and ßNrxn have been reported as well for all three Nrxn genes (reviewed in [106,108]). Assuming that the mutations reported in the literature are causative, it is interesting to note that the severity of clinical symptoms does not appear to correspond to any particular isoform (αNrxn, ßNrxn, or their combination) or gene (Nrxn1-3) that is affected.

While no ASD-related behavioral analyses has been published thus far for ßNrxn knockout mice, relevant abnormalities, such as increased repetitive behavior and impaired social interaction, were observed in a transgenic mouse model overexpressing a defective Nrxn1ß construct lacking c-terminal sequences (Nrx1ßΔC) in adult mice [123]. Most importantly, alteration of dendritic spines is generally a hallmark of mouse models of neurodevelopmental disorders [16,17,18,124] and is reported here for ßNrxn-deficient neurons and brains. While most disease models reveal an up- or downregulated overall spine density [10,17], the combination of an increase in mature mushroom-type spines with an unchanged overall spine density (Figure 3 and Figure 4) and more perforated PSDs (Figure 6), as seen here, is relatively rare. Exactly this combination of structural changes, in addition to reduced LTP and EPSPs, was also observed in a mouse model of hippocampal demyelination when studying cognitive decline in patients with multiple sclerosis [125]. This is an interesting result because it confirms that under pathological conditions, dysfunctional spinous synapses can have more—not less—mature spines with more perforated PSDs. Such findings are in contrast to the general opinion that a mature mushroom spine morphology reflects intact, strong presynaptic function [8,65,95]. While mouse models of diseases provide essential insights into possible pathomechanisms, there are also caveats. An important lesson could recently be learned from critical tests of the prominent excitatory-to-inhibitory synaptic conductance idea or the E-I ratio hypothesis of ASD [126,127]. Although many ASD mouse models show an increased E-I ratio, this might rather reflect a compensation to stabilize synaptic depolarization than lead to hyperexcitability at the level of neural circuitry or social impairment in patients [128,129]. Thus, understanding how specific alterations at synapses—possibly via compensatory processes—translate to the systemic level in ASD patients remains a challenge for the future.

## 4. Materials and Methods

### 4.1. Animals

The mouse strains used in this study were previously described in [27,31,47]. Briefly, conditional deletion of all βNrxn variants was achieved via transduction with Cre-recombinase-expressing lentivirus in neuronal cell cultures from a triple-βNrxn-floxed mouse strain (βNrxn KI; available from JAX Labs as B6;129-Nx1TM 2 Sud Nx2TM 2 Sud Nx3TM 2 Sud/J, RRID:IMSR_JAX:008416). The constitutive triple-βNrxn-knockout mice (βNrxn TKO) were created from the same line by crossing to a transgenic Cre recombinase deleter strain (B6.FVB-Tg (Ella-Cre) C5379Lmgd/J; RRID:IMSR_JAX:003724). Conditional and constitutive KO strains were subsequently bred into a C57BL/6J background (RRID:IMSR_JAX:000664) over multiple generations. Mice were maintained at the central animal facility in Münster under standard housing conditions with food and water available ad libitum on a 12 h light/dark cycle. All animal experiments were performed at the University of Münster according to government regulations for animal welfare.

### 4.2. Analysis of βNrxn-Deficient Neurons in Culture

#### 4.2.1. Neuronal Cell Culture

Primary hippocampal neurons were prepared from timed-pregnant βNrxn KI and βTKO mice, plated on poly-L-lysine-coated glass coverslips, cocultured on a 70–80% confluent monolayer of mouse astrocytes grown in 12-well plates, and maintained at 37 °C in an atmosphere of 95% air and 5% CO_2_, as described in [44,47]. All analyses were performed on the culture at 19days in vitro (DIVs).

#### 4.2.2. Lentivirus Production and Transduction

The generation of lentivirus particles and the subsequent infection protocol have been described in detail before [31,47]. In this study, βNrxn KI neurons in culture were transduced by two lentivirus vectors: (i) active Cre-recombinase-expressing virus particles to delete all βNrxn or (ii) inactive Cre^mut^ virus for controls. Lentivirus particles were added to neuronal cultures at DIV5 for 3 days. Infection efficiency was routinely checked by using the nuclear expression of EGFP, which was fused to Cre/Cre^mut^ constructs, and only preparations with an efficiency above 95% were included in our experiments. In addition, fluorescent labeling of MAP2, a cytoskeleton marker in neuronal cell bodies and dendrites, was performed in every experiment to verify healthy cell cultures and to exclude cells with compromised architectures.

#### 4.2.3. Transfection of Cell Cultures

To visualize processes and dendritic spines, βTKO neurons were transfected with pSyn5-t-dimer2-RFP cDNA (T. Oertner, Basel, Switzerland) at DIV14. In brief, mixtures of 2 µg Lipofectamine 2000 (Fisherscientific, Waltham, MA, USA, Cat #11668019) with 100 µL of neurobasal medium (NBM) and 1 µg of pSyn5-t-dimer2-RFP DNA with 100 µL of NBM were combined and incubated for 20 min. The mix was then added dropwise to the neuronal cell culture and incubated for 30 min. Before being transferred back to the astrocyte plate, the neuronal coverslips were washed twice with NBM to remove the transfection mixture. Cells were used 5 days after transfection for immunofluorescence studies.

#### 4.2.4. Immunocytochemistry of Cell Cultures

For chromogranin A labeling, we used polyclonal rabbit anti-ChrgA primary antibody (1:500; Synaptic Systems, Göttingen, Germany; Cat #259003, RRID:AB_2619972) and goat anti-rabbit conjugated to Cy3 as a secondary antibody (1:500; Jackson ImmunoResearch Labs, Ely, UK; Cat #111-165-003, RRID: AB_2338000). For MAP2 labeling, the primary antibody used was polyclonal chicken anti-MAP2 (1:5000; Abcam, Boston, MA, USA; Cat #ab5392, RRID: AB_2138153), along with goat anti-chicken as a secondary antibody conjugated to Alexa fluor 647 (1:500; Thermo Fisher Scientific, Darmstadt, Germany; Cat #A21449, RRID: AB_2535866).

#### 4.2.5. Fluorescent Microscopy of Cell Cultures

Coverslips with primary neurons were dipped in PBS with 4% sucrose and subsequently fixed in 4% paraformaldehyde with 4% sucrose in PBS for 10 min. After washing, they were incubated in a blocking solution (0.3% TritonX-100 and 5% NGS in PBS) for 30 min. Primary antibody staining was performed on a shaker at RT for 1 h. Neurons were washed and incubated with the secondary fluorescent antibody at RT for 1 h in the dark. Both antibody solutions contained 5% NGS to prevent unspecific binding. Finally, after washing again in PBS, the coverslips were mounted on slides with Dako fluorescent mounting medium. Images were acquired with a x63 oil immersion objective in a VisiScope cell analyzer/Zeiss Axio Imager.Z2 fluorescence microscope (Zeiss, Oberkochen, Germany) equipped with the VisiView software (version 4.0.0.16; Visitron Systems, Puchheim, Germany), and they were analyzed with the ImageJ software (NIH Image, version 2.14.0/1.54f, RRID: SCR_003073).

#### 4.2.6. Analysis of ChrgA-Stained Axons

The analysis of the anti-ChrgA fluorescent intensity of conditional βNrxn KO and constitutive βNrxn TKO neurons and their respective controls was performed by an investigator blinded to the genotype. ChrgA-labeled parts of an axon were outlined with rectangular measuring windows with a width of 5 µm. A maximum of 100 μm of ChrgA-positive axonal length per image was analyzed, and an average value was calculated. Similar axon lengths were studied for each genotype, including approximately 2800 μm for the constitutive βNrxn TKO and approximately 1600 μm for conditional βNrxn KO plus the same lengths from their respective control cultures. Three sets of independent neuronal cultures (*n* = 3) from each genotype were included in this analysis.

#### 4.2.7. Analysis of RFP-Labeled Dendrites and Spines

The length of all spines along a dendritic segment and the length of the dendritic segment itself were measured in conditional βNrxn KO and constitutive βNrxn TKO neurons and in their respective controls. As outlined in Figure 2, spines longer than 2 μm were marked with a yellow segmented line, and spines shorter than 2 μm were marked with a turquoise point. Branched spines were counted as one spine, and only the longer branch was measured and included in the quantification. For spine densities, the overall number of spines, the number of spines longer than 2 µm, and the number of spines shorter than 2 μm were divided by the length of dendritic segments analyzed. In summary, we examined a total dendritic length of 6720 μm with 4601 spines for constitutive βNrxn TKO plus 5010 μm with 3160 spines for their controls and 3450 μm with 2338 spines for conditional βNrxn KO neurons plus 3490 μm with 2295 spines for their controls.

### 4.3. Electron Microscopic Analysis of βNrxn-Deficient Brain Tissue

#### 4.3.1. Preparation of Mouse Hippocampi

The protocols for the transcardial perfusion, sample preparation, and embedding of brain tissue from constitutive βNrxn TKO and control mice for transmission electron microscopy were recently described in detail [47]. For this study, we used three mice per genotype and trimmed the resin-embedded hippocampal samples to include the stratum radiatum of CA1 regions. Ultrathin sections with a thickness of approximately 70 nm were mounted on Formvar-coated copper grids and examined on a LIBRA 120 transmission electron microscope (Zeiss, Oberkochen, Germany) at 80 kV.

#### 4.3.2. Three-Dimensional Reconstructions

For 3D reconstructions of dendritic spines, we obtained serial sections from small sharp-edged tissue blocks using a diamond trimming knife (Trim 45 T3516; Diatome, Nidau, Switzerland). A continuous band of about 25–30 serial sections (approximately 50 nm thick with dimensions of 0.1 × 1.0 µm) was cut, picked up with a droplet of distilled H_2_O on an uncoated copper slot grid, and transferred to the droplets. Sections were contrasted with a saturated uranyl acetate solution, followed by a lead citrate solution. Washing was achieved by transferring the grid from one drop to another (15 drops in total). The grid was then put on a Formvar-foil-coated hole of a plastic object slide (Science Services, Munich, Germany; Cat #E71891-10, foil), with the foil being thicker according to the manufacturer’s instructions. The slide/grid sandwich was dried overnight. The next day, the foil was perforated around the grid with forceps, and the grid was ready to use for TEM. Serial sections through a spine were imaged at the same position with ×6300 magnification until the spine disappeared.

#### 4.3.3. Ultrastructural Image Analysis

To study the ultrastructure of mushroom spines (total number, area of spine heads, length of spine necks) and the number of synapses with perforated PSDs, we obtained images with a magnification of ×1260 (or ×6300 for illustrative purposes) using the TEM morphometry software (version 1.2.9.112; Tröndle, Moorenweis, Germany). The following criteria had to be fulfilled to count for a mushroom spine: Firstly, the dimension of the head width (Figure 7A, green area) was at least twice the diameter of the neck (Figure 7A, red area). Secondly, the total length of the spine was not longer than 2 µm, and thirdly, a clear connection to a dendrite had to be present. The latter was also important for the measurement of the length of the neck (Figure 7A, yellow line). A total area of 13,760 µm^2^ was studied for constitutive βNrxn TKO and similarly for the control mice to count the mushroom spine numbers (739 in TKO, 498 in the control) and the numbers of spines with perforated PSDs (163 in TKO, 133 in the control). Fifty spines from each experiment (150 in total) were further analyzed for their head area and neck length. In addition, the numbers of asymmetric synapses were counted in images taken at a magnification of ×4000 with a total investigated area of 1840 µm^2^ for each genotype; 1027 synapses were identified in TKO animals, and 1013 were identified in the control animals. To count as an asymmetric Gray type 1 synapse [64,130], at least three presynaptic vesicles, a visible synaptic cleft, and prominent postsynaptic density had to be present.

### 4.4. Software and Statistics

#### 4.4.1. Three-Dimensional Reconstruction

For the alignment of serial sections, an algorithm was programmed in MATLAB. Dendritic spines of interest were then segmented using the TrakEM2 plugin of ImageJ. After assembly into 3D structures, each spine was interactively edited with the remesh modifier of the Blender program (Version 3.4.1; RRID:SCR_008606). To reconcile the smoothing of the surface and for the preservation of relevant details, the following settings were chosen for the voxel size: dendrite, 0.1 m; spine, 0.05 m; PSD, 0.02 m. Ten reconstructed spines of each genotype are presented in Figure 4.

#### 4.4.2. Statistical Analysis

All statistical analysis was performed with the Prism Software (Version 7.0e, GraphPad; RRID:SCR_002798). The data in the figures are shown ±SEM for the KO and control samples. To assess the statistical significance between the two groups, we used a two-tailed unpaired Student’s *t*-test with Welch’s correction. Significance differences are indicated in detail in the corresponding figure legends.

## Figures and Tables

**Figure 1 ijms-25-01285-f001:**
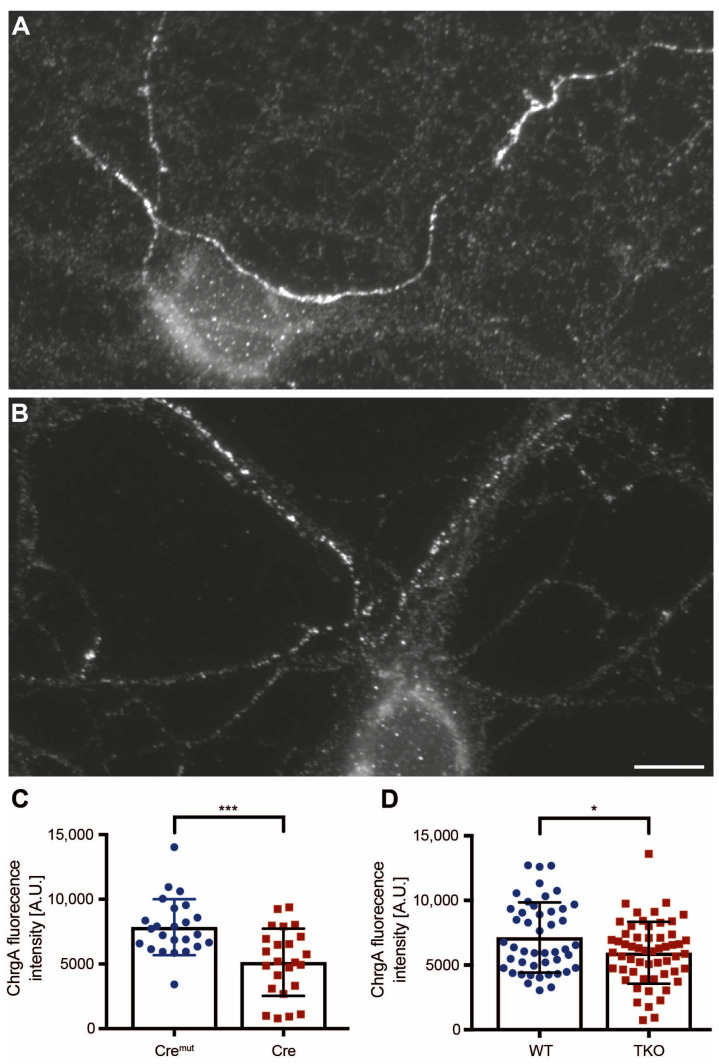
Reductions in chromogranin A (ChrgA) levels in conditional and constitutive ßNrxn KO neurons. Representative images for ChrgA immunofluorescence labeling of primary hippocampal neurons from control (**A**) and constitutive ßNrxn TKO (**B**) mice. Scale bar (for (**A**,**B**)), 10 μm. (**C**) Quantification of mean ChrgA fluorescence intensity in floxed ßNrxn KI neurons transduced with lentivirus expressing either inactive (Cre^mut^) or active Cre recombinase. (**D**) Comparison of mean fluorescence intensity in neurons from constitutive ßNrxn TKO neurons and their controls (WT), confirming reduction of ChrgA-positive DCVs in the absence of ßNrxn. Data are shown as mean ± SEM and are based on N = number of axonal segments (Cre = 24, Cre^mut^ = 24; WT = 46, TKO = 56) from three independent cultures per genotype and mouse line; two-sided unpaired *t*-test with significance levels indicated as * *p* < 0.05, *** *p* < 0.001.

**Figure 2 ijms-25-01285-f002:**
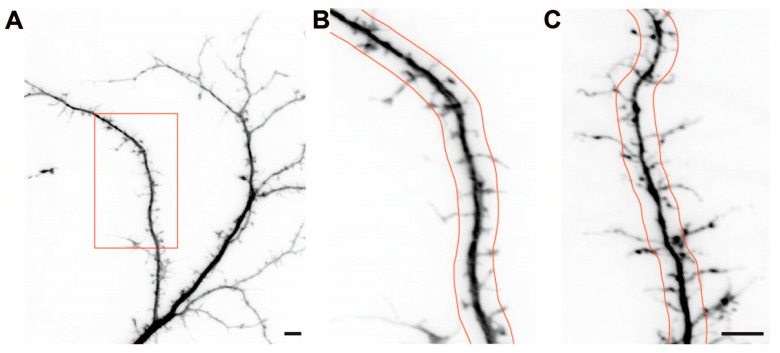
Morphology of dendritic protrusions in cultured βNrxn-deficient neurons. Representative images of dendrites and their protrusions of primary hippocampal neurons from the WT control (**A**,**B**) and constitutive ßNrxn TKO (**C**) mice. About 10–15 neurons per coverslip were transfected with pSyn5-t-dimer2-RFP at DIV14 and imaged at DIV18/19. Fluorescent signals from images with lower (**A**) and higher (**B**,**C**) magnification were inverted for better contrast; the inset in (**A**) corresponds to (**B**). Dendritic protrusions were classified as filopodia (length > 2 μm) and all other spine types (length < 2 μm); red lines visualize the 2 μm border. Scale bar in (**A**), 5 μm; scale bar in (**C**) (for (**B**,**C**)), 5 μm.

**Figure 3 ijms-25-01285-f003:**
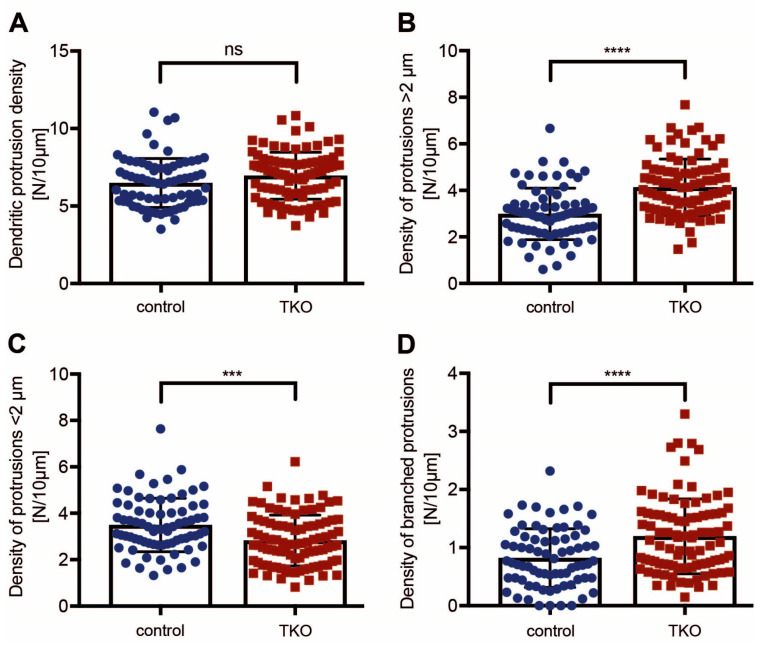
Increased proportion of longer and branched dendritic protrusions in ßNrxn-deficient neurons. (**A**) The overall number of protrusions per 10 μm of dendritic length did not significantly differ between the control and ßNrxn TKO neurons. The number of filopodial protrusions longer than 2 μm was increased in mutant neurons (**B**), while the number of spines shorter than 2 μm was decreased (**C**). (**D**) The number of branched spines was augmented in ßNrxn TKO neurons. Data are shown as mean ± SEM and are based on N = number of dendritic segments (control = 73, TKO = 93) sampled from 30–36 mRFP-transfected neurons on nine coverslips from three independent cultures per genotype; two-sided unpaired *t*-test with *** *p* < 0.001, **** *p* < 0.0001, and ns = non-significant.

**Figure 4 ijms-25-01285-f004:**
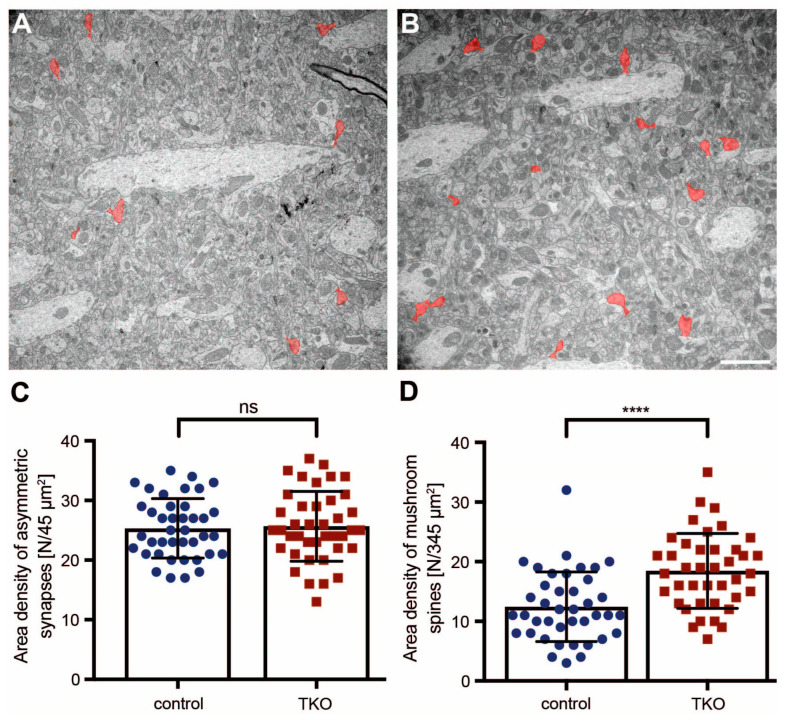
Increase in the number of mushroom spines in ßNrxn-deficient mice. Representative electron microscopic images of the stratum radiatum of hippocampal CA1 regions from control (**A**) and ßNrxn TKO (**B**) mice. Images were taken at a primary magnification of ×1260 on a Zeiss LIBRA120 TEM; mushroom spines are colored in red; scale bar (for (**A**,**B**)), 2 μm. (**C**) The area density of asymmetric type 1 synapses in the control and ßNrxn TKO samples was unchanged. (**D**) The area density of mushroom spines was increased in ßNrxn TKO compared to the control. Refer to Section 4.3.3 for ultrastructural criteria of asymmetric synapses and mushroom spines. Data are shown as mean ± SEM and are based on N = number of neuropil areas analyzed ((**C**), control = 40 areas of 45 μm^2^, TKO = 40; (**D**), control = 40 areas of 345 μm^2^, TKO = 40 from five animals per genotype; two-sided unpaired *t*-test with **** *p* < 0.0001, and ns = non-significant.

**Figure 5 ijms-25-01285-f005:**
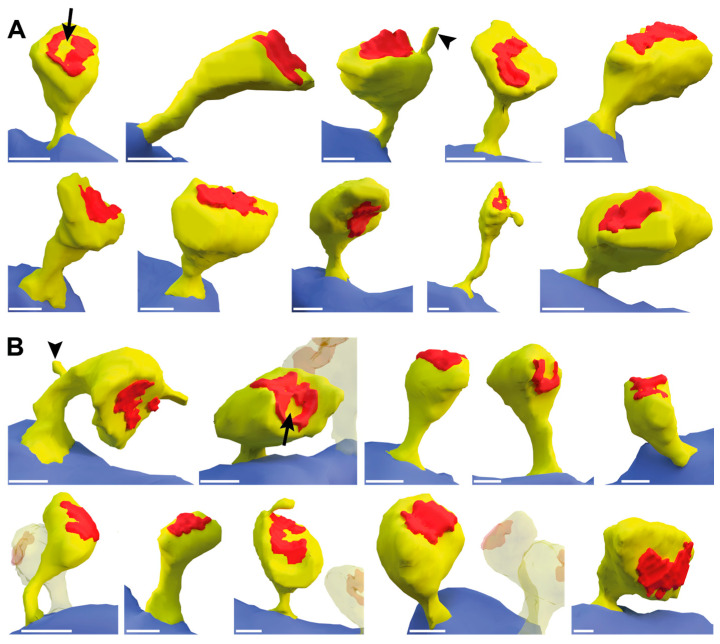
Three-dimensional reconstructions of dendritic spines. Mushroom-type spines were reconstructed in 3D from serial sections prepared for electron microscopy from WT control (**A**) and ßNrxn TKO (**B**) samples of the stratum radiatum in hippocampal CA1 regions. Relevant ultrastructural features were interactively outlined on all aligned images and color-coded as dendritic processes (blue), spinous protrusions with a head and neck (yellow), and postsynaptic densities (PSDs) (red). In addition, arrows point to examples of PSDs with electron-lucent regions, commonly referred to as perforated PSDs [67]; arrowheads show examples of spinules. Note that the samples were adjusted to a comparable size; all scale bars represent 0.2 μm.

**Figure 6 ijms-25-01285-f006:**
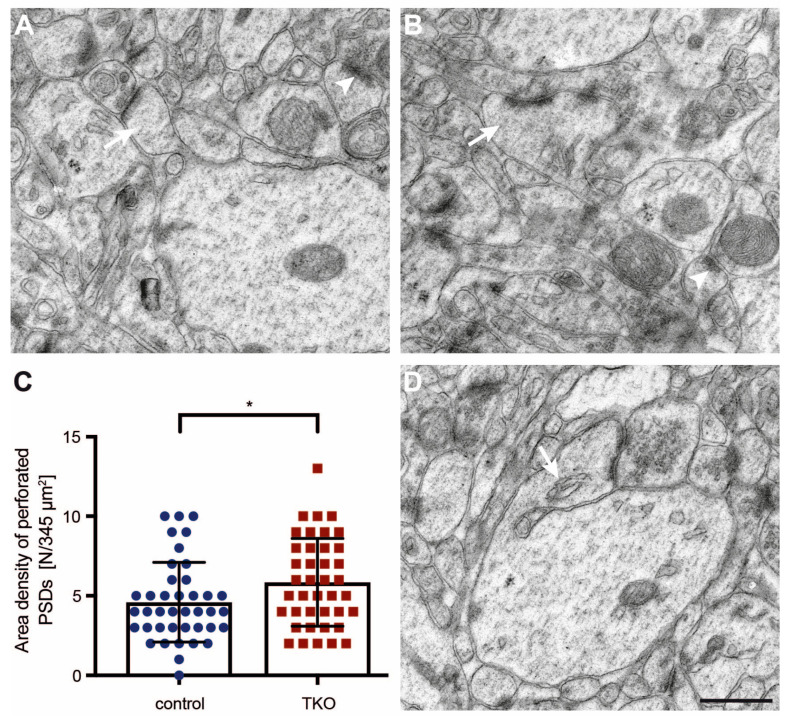
Dendritic spines with perforated PSDs occurred more frequently in ßNrxn-deficient mice. (**A**) Characteristic mushroom spine with a macular PSD (white arrow) extending from a dendritic process in the stratum radiatum of the CA1 hippocampus. (**B**) Mushroom-type spine containing a perforated PSD (white arrow) from the same region as that in (**A**). The arrowheads in (**A**,**B**) point to additional examples of asymmetric type 1 synapses. (**C**) The area density of mushroom spines with a perforated PSD was increased in ßNrxn TKO compared to the control brains. Data are shown as mean ± SEM and are based on N = number of neuropil areas analyzed (control = 40 areas of 345 μm^2^, TKO = 40) from five animals per genotype; two-sided unpaired *t*-test with * *p* < 0.05. (**D**) Mushroom spines containing a spine apparatus—indicated by the white arrow—were present in about 15% of both the control and TKO samples. The images in (**A**,**B**,**D**) were taken at a primary magnification of ×6300 on a Zeiss LIBRA120 TEM; scale bar (for **A**,**B**,**D**), 0.5 μm.

**Figure 7 ijms-25-01285-f007:**
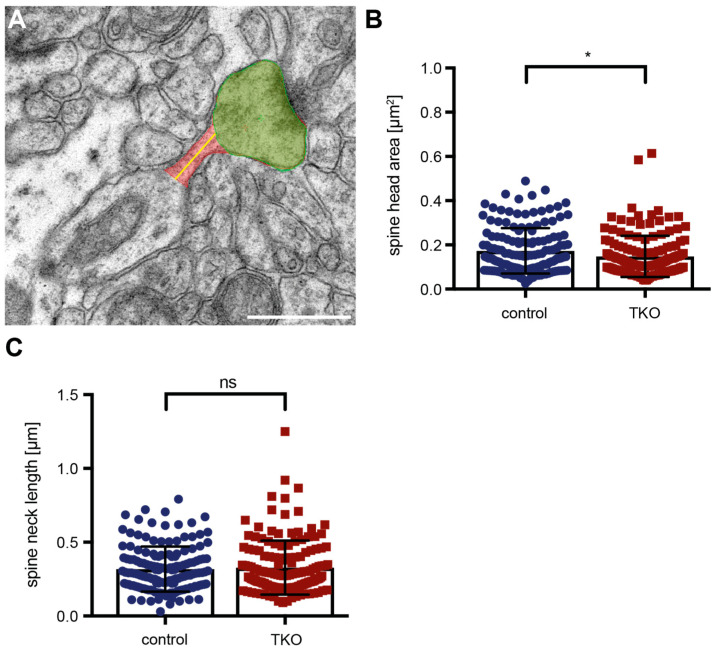
Head and neck dimensions of mushroom spines in ßNrxn TKO mice. (**A**) Representative image of a mushroom spine (head area, green; neck, red; neck length, yellow line) extending from a dendritic process in the stratum radiatum of the CA1 hippocampus. The image was taken at a primary magnification of ×6300; scale bar, 0.5 μm. (**B**) Average head area and (**C**) neck length of mushroom spines in ßNrxn TKO neurons compared to the control neurons. Data are shown as mean ± SEM and are based on N = number of mushroom spines (control = 150, TKO = 150) from three animals per genotype; two-sided unpaired *t*-test with * *p* < 0.05, and ns = non-significant.

## Data Availability

Data are contained within the article, and the raw data presented in this study are available upon request from the corresponding author.

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
