# Peer review of "Distinct Alterations in Dendritic Spine Morphology in the Absence of β-Neurexins"

_ijms, 2024, doi:10.3390/ijms25021285_

Round 1
Reviewer 1 Report
Comments and Suggestions for Authorsβ-Neurexins represent shorter isoforms within the Neurexin family, a group of presynaptic cell adhesion proteins. Compared to their longer counterparts, α-Neurexins, there is limited understanding of the roles played by β-Neurexins. In this paper, the authors have employed a β-Neurexin-1, 2, 3 triple knockout mouse model to analyze the impact of β-Neurexin deletion on synapse morphology, both in culture and in vivo. Utilizing techniques such as immunocytochemistry and electron microscopy, they have identified that the absence of β-Neurexins influences the localization of dense core vesicles and alters the morphology of dendritic spines. Given the current paucity of information regarding β-Neurexin, these findings constitute a significant contribution to the field of molecular neurobiology. The manuscript is well-crafted, effectively integrating their findings with previous research in both the results and discussion sections. In my opinion, this paper meets the high standards required for publication in the International Journal of Molecular Sciences (IJMS).
Author Response
Thanks for the very positive assessment of our manuscript.
Reviewer 2 Report
Comments and Suggestions for Authors
This study provided in-depth analyses using EM and immunohistochemistry analyses from beta-neurexin triple knockout mice to examine any alterations in dendritic spines of cultured neurons and slice preparations. The study concluded that beta-neurexins are required for normal dendritic spine development. Data are of overall high-quality and I support the publication of this study. However, authors need to provide exact "n" numbers for all figures and precisely indicate whether the number of mice, sections, neurons, or spines was used for statistical assessment. This should conform to standard practice in the field.
Author Response
Thanks for the overall positive assessment of our manuscript. We have now resolved the remaining issue with the description of methods by including in each figure legend the exact N numbers used for the statistics and how many cultures, mice, etc. per genotype these numbers were derived from. We also amended the M&M section to contain more details on these issues.
Reviewer 3 Report
Comments and Suggestions for Authors
The manuscript by Mohrmann et al. analyzes the morphological effect of β-neurexins on dendritic spines. β-neurexins are definitely absent from postsynaptic membrane. Therefore, all possible effects on the dendritic spine morphology are indirect, either through extracellular matrix+neuroligin-PSD-95, or functionally, by altered neurotransmitter release. This manuscript does not try to clarify this issue. It just focuses on postsynaptic structure morphology, no matter what causes them. Although the manuscript is interesting and important for the field, there are several issues, listed below.
Line 31 … they constitute an independently operating compartment of the postsynaptic neuron
This is an unjustifiably categorical statement. The degree of independence of the spine from the rest of the neuron, without specifying details, misleads the reader.
Line 37 Since spines undergo adaptive changes in response to stimuli, it was concluded that spine morphology and their function are mutually dependent
I do not see how adaptive changes indicate a relationship between morphology and function
Lines 39-40 For example, the normally round and flat surface of spines that faces the presynaptic terminal can assume a concave morphology following plasticity-inducing signals
I have a hard time imagining a round and a flat surface at the same time.
Line 48 LNS domain
Apart from the most common abbreviations, like DNA, others need to be explained.
Lines 81-84 At a time when replicability of exactly comparable experimental conditions is a topic of great concern in science, we decided to study dendritic spine structure in two different βNrxn deletion models and compared their phenotype in neuronal cell culture as well as brain tissue.
I don’t understand the logic and meaning of this sentence: what does the reproducibility of scientific results have to do with it and how does it relate to the comparison of different models?
Lines 75-121 It remains unclear to me the logical correspondence of section 2.1 to the title of the article. If the manuscript is titled: “Distinct alterations of dendritic spine morphology in the absence of β-neurexins,” then what does the axon data have to do with it? The title should be consistent with the content of the article and not contradict it.
Lines 122-171 I have a number of questions and concerns regarding the data presented in Section 2.2 and in Figures 2 and 3. These data are especially important because they, and not electron micrographs, allow a reader to assess the condition of the tissue. Particularly:
1. Hippocampal culture shown on 2A looks an unusual as 19-day in vitro preparation. This particular example looks to be taken from immature or deafferented/poorly afferented neuron.
2. The quality of the image does not allow judging about the numbers and types of protrusions, most of which are clearly filopodia.
3. On the example shown on 2A, I can only see 1 or 2 mushroom spines per entire 35 um segment. The rest are either small stubby or filopodia. I can speculate that transfection, performed quite late, at DVI 14 catches mostly immature cells.
4. Without correct control, the analysis of ßNrxn TKO is not applicable. Most of the protrusions shown on 2B are 5-10 um long. What kind of spines are these? Filopodia are unstable transient structures, changing their lengths/shapes in second-long timescale. The reason for their categorization is questionable. It is very unusual to call such structures “spines”. They are definitely not. From the only example shown, I can reveal about 1 spine per 10 um, which is not consistent with Anderson et al., as authors claim.
5. Figure 3 caption: it is unclear how many cells per culture and segments per cell have been analyzed?
Figure 4 What is the scale bar for 4A&B? If the field of 45 um2 is given, then the numbers are not consistent with averages on 4D (examples look taken from the top edge for TKO and bottom edge for control). Please, show more representative examples.
Lines 200-214 Since the procedure for identifying asymmetric synapses and mushroom-shaped dendritic spines is key to this section (and to the paper, overall), it should be accompanied by convincing illustrations of identification from electron micrographs.
Figure 5 A big issue of this set is extremely different scales (up to 4-time difference), which highly disturb to evaluate visually the data shown.
Figure 6 Data with spine apparatus is very interesting. It is very likely that the presence/absence of spine apparatus is the key point of the difference between KO and control. I highly suggest authors summarizing these data rather than just mentioning SA.
Comments on the Quality of English LanguageEnglish sounds a bit heavy, some sentences are too long and hard for understanding.
Author Response
Thanks for the detailed and constructive assessment of our manuscript. We believe that we could clarify or resolve all remaining issues by adding new experimental data, improving representative images, modifying figure panels, and rephrasing parts of the text. We have outlined these changes in detail in the rebuttal letter attached below.
